# Preparation of Encapsulated Breakers for Polymer Gels and Evaluation of Their Properties

**DOI:** 10.3390/gels9050387

**Published:** 2023-05-08

**Authors:** Kaihe Lv, Guodong Zhang, Yingrui Bai, Jingbin Yang

**Affiliations:** 1Department of Petroleum Engineering, China University of Petroleum (East China), Qingdao 266580, China; zhangguodong1106@126.com (G.Z.);; 2Key Laboratory of Unconventional Oil & Gas, Development Ministry of Education, Qingdao 266580, China

**Keywords:** encapsulated breaker, in situ polymerization, gel temporary plugging, delayed gel breaking

## Abstract

A common problem associated with conventional gel breakers is that they can cause a premature reduction in gel viscosity at high temperatures. To address this, a urea-formaldehyde (UF) resin and sulfamic acid (SA) encapsulated polymer gel breaker was prepared via in situ polymerization with UF as the capsule coat and SA as the capsule core; this breaker was able to withstand temperatures of up to 120–140 °C. The encapsulated breaker was characterized using scanning electron microscopy (SEM), infrared spectroscopy (FT-IR), and thermogravimetric (TG) analysis. Meanwhile, the dispersing effects of various emulsifiers on the capsule core, and the encapsulation rate and electrical conductivity of the encapsulated breaker were tested. The gel-breaking performance of the encapsulated breaker was evaluated at different temperatures and dose conditions via simulated core experiments. The results confirm the successful encapsulation of SA in UF and also highlight the slow-release properties of the encapsulated breaker. From experimentation, the optimal preparation conditions were determined to be a molar ratio between urea and formaldehyde (n_urea_:n_formaldehyde_) of 1:1.8 for the capsule coat, a pH of 8, a temperature of 75 °C, and the utilization of Span 80/SDBS as the compound emulsifier; the resulting encapsulated breaker exhibited significantly improved gel-breaking performance (gel breaking delayed for 9 days at 130 °C). The optimum preparation conditions determined in the study can be used in industrial production, and there are no potential safety and environmental concerns.

## 1. Introduction

Malignant lost circulation in fractured formations is one of the most difficult-to-manage downhole complications; this also gives rise to other issues (i.e., loss of drilling fluid) that can result in serious economic losses and restrict the progress of oil and gas exploration and development [1]. Temporary plugging agents based on polymer gels are widely used for lost circulation treatment in fractured formations because they are not restricted by lost circulation channels [2]. However, these plugging agents suffer from uncontrollable gel-breaking time and can result in the secondary contamination of the reservoir [3]. Gel breakers are one of the main additives in temporary plugging materials [4]; they can break the gel molecular chains and reduce the relative molecular mass of the polymer molecule, thus promoting fluid flowback and alleviating the secondary contamination of the reservoir. With the deepening of oilfield exploitation, the performance of gel breakers is affected by the pH and temperature conditions of the environment, which limits their wide applicability [5]. Conventional gel breakers can cause a premature reduction in the viscosity of the gels at high temperatures, resulting in poor temporary plugging performance; thus, they are not suitable for use in high-temperature applications [6]. The gel breakers used in the temporary plugging process must be able to maintain gel viscosity and alleviate damage to the oil and gas reservoir; to combat this, encapsulated breakers can be utilized [7]. As shown in Figure 1, the encapsulated breaker is mixed with a gel solution and subsequently injected into the ground to form gels, reducing engineering costs. The encapsulation technology is a method of enclosing dispersible solids, liquids, or gases with polymer materials to form a core-shell structure that protects the capsule core from the impact of external environments, such as temperature and oxygen, and controls the release of the active ingredient. In the petroleum industry, the temperature resistance of other alternatives is only 50–80 °C. In this study, we prepared an encapsulated breaker with a core-shell structure and a temperature resistance of up to 120–140 °C via in situ polymerization, using urea-formaldehyde (UF) resin as the capsule coat and sulfamic acid (SA) as the capsule core. The microstructure, encapsulation rate, and stability of the encapsulated breaker were analyzed, and the gel-breaking performance of the encapsulated breaker in a high temperature environment was studied. Compared with the existing encapsulated breaker technology, the novelty of this study is to improve the temperature resistance of encapsulated breakers. The UF/SA encapsulated breaker has the advantages of low cost, high efficiency, and sustainability. In the future, the temperature resistance of encapsulated breakers should be improved by optimizing the preparation method.

## 2. Results and Discussion

### 2.1. Result Analysis

#### 2.1.1. TEM Characterization of Capsule Core Emulsions and SEM Characterization of Encapsulated Breaker

Emulsifiers play a critical role in the stability of emulsions due to their tendency to be adsorbed at the oil–water interface. These surface-active agents are responsible for lowering the interfacial tension and facilitating the formation of emulsion droplets [8]. The type of emulsifier used in the preparation process affects the stability and dispersion of the oil–water system and is key to the formation of the encapsulated breaker [9]. In this study, TEM characterizations were carried out on emulsions formed with SDBS, Span 80, and the Span 80/SDBS compound emulsifier (Figure 2). Using SDBS or Span 80 as a single emulsifier results in the aggregation of the capsule cores, confirming the instability of the emulsion system. On the other hand, the Span 80/SDBS compound emulsifier results in a significant improvement in the dispersion of the capsule cores with little to no aggregation observed. This can be attributed to the interaction between SDBS (anionic surfactant with strong hydrophilicity) and the hydrophilic groups in Span 80 that lowers the oil–water interfacial tension, reducing aggregation [10]. Figure 3 shows the SEM images of the encapsulated breakers prepared with the Span 80/SDBS compound emulsifier. Urea-formaldehyde resin prepared from formaldehyde solution and urea in different proportions will cause the encapsulated breakers with different structures. The encapsulated breaker prepared with a n_urea_:n_formaldehyde_ ratio of 1:1.8 was found to be spherical with a smooth capsule coat, while that with a n_urea_:n_formaldehyde_ ratio of 1:1.7 was found to be rough and wrinkled. This confirms that the UF prepolymer can be deposited on the surface of uniformly dispersed SA capsule cores, forming encapsulated breakers with a relatively regular spherical shape.

#### 2.1.2. IR Spectroscopy Analysis

IR spectroscopy analysis was carried out to determine whether the prepared encapsulated breaker was formed from enclosing the SA capsule core with the UF resin capsule coat [11]. As shown in Figure 4, the absorption peaks attributed to the N-H and O-H groups overlap with each other to form an intensified absorption peak at 3402 cm^−1^; the stretching vibration absorption peak of C=O in the amide bond is wide at 1650 cm^−1^, and the -OH absorption peak at 1430 cm^−1^ is considerably weaker. The stretching vibration absorption peaks of C-N and N-H in the amide bond can be observed at 1390 cm^−1^. This confirms the formation of the UF resin capsule coat. Additionally, the peaks at 1280, 1190, 1050, and 860 cm^−1^ can be attributed to the stretching vibration of -SO₃H. The peaks of urea-formaldehyde resin appear first, and those of sulfamic acid appear later. In summary, the UF resin capsule coat has successfully enclosed the SA capsule core to form the encapsulated breaker with a core-shell structure.

#### 2.1.3. TG Test and Analysis

The TG test is an important way to analyze material stability [12]. The TG curve of the encapsulated breaker is shown in Figure 5. The weight loss in the temperature range of 0–100 °C is low and is caused by the evaporation of small molecules, such as water and free formaldehyde, from the surface of the capsule coat [13]. The weight loss increases significantly between 120 °C and 140 °C, which can be attributed to the rupture and rapid decomposition of the capsule coat, resulting in the rapid release of the capsule core. The weight loss then increases at a slower rate between 200 °C and 600 °C, featured by the further decomposition of the capsule coat. From the TG curve of the UF resin, it can be observed that the UF resin loses weight slowly between 0 °C and 300 °C; the rate of weight loss reaches the maximum between 310 °C and 350 °C. At this stage, the UF resin decomposes rapidly. When the temperature is increased to 400 °C and beyond, the UF resin continues to decompose and the nitrogen residue in the resin continues to decrease. These results prove that the UF prepolymer was able to successfully enclose SA, and the encapsulated breaker can be safely stored at room temperature and used for gel-breaking construction between 120 °C and 140 °C. From the TG curves, it can be inferred that the temperature resistance of the encapsulated breaker can be improved by increasing the thickness of the capsule coat. In summary, the temperature resistance of the encapsulated breaker can be adjusted to accommodate different construction requirements and environments, so as to achieve controllable gel-breaking time.

### 2.2. Encapsulation Rate of Encapsulated Breaker

The encapsulation rates of the encapsulated breakers prepared via in situ polymerization were found to be 60% or higher in all groups, as shown in Table 1, which indicates that the UF resin capsule coat successfully enclosed the capsule core SA. Because the dispersion of the capsule core in different emulsifiers is different, the encapsulation rate of the encapsulated breaker is different. Other conditions being the same, the better the dispersibility, the higher the package rate. Given a constant n_urea_:n_formaldehyde_ ratio, the encapsulation rate of the encapsulated breaker prepared with the Span 80/SDBS compound emulsifier was found to be higher than that of encapsulated breakers prepared with the Span 80 or SDBS single emulsifier. This indicates that the compound emulsifier lowers the aggregation probability of the capsule cores during the formation of the capsule structure and enhances the stability of the emulsion during the mixing process [14], thus increasing the deposition rate of the prepolymer and improving the encapsulation rate. The encapsulation rate of the UF/SA encapsulated breaker in group 6 was found to be the highest, which was prepared under the conditions of n_urea_:n_formaldehyde_ at 1:1.8, pH at 8, temperature at 75 °C, and with the Span 80/SDBS compound emulsifier.

### 2.3. Electrical Conductivity of Encapsulated Breaker

The dissolution of the SA capsule core in water leads to changes in the conductivity of the solution, which means that the change in solution conductivity is directly correlated to the SA concentration in the solution [15]. Therefore, the release of the capsule core from the encapsulated breaker can be studied by measuring the solution conductivity at different temperatures. As shown in Figure 6, the conductivity of the SA solution increases with time; notably, the rate of increase in conductivity appeared to be higher in the first 12 h [16]. This can be attributed to the rapid dissolution of SA in water that produces a large number of sulfate ions. After 12 h, the rate of the dissolution of SA decreases, which is followed by a corresponding decrease in the increasing rate of the conductivity. The conductivity of the encapsulated breaker solution was found to always be lower than that of the SA solution under the same temperature at the same time points. Therefore, when not damaged by external forces, the UF resin capsule coat enables the slow release of the SA capsule core. The conductivity of the SA solution at the same time point increases with the increasing temperature; this can also be observed in the encapsulated breaker solution, which shows that the capsule coat structure breaks more easily at high temperatures [17], leading to an increased release rate of the capsule core and affecting the stability of the encapsulated breaker.

### 2.4. Gel-Breaking Performance of Encapsulated Breaker

#### 2.4.1. Effect of Temperature on Gel-Breaking Performance

To study the effect of temperature on gel-breaking time, a liquid was formulated according to Section 4.3.1. The change in gel viscosity with time at different temperatures was measured using a rheometer. According to the results shown in Figure 7, it can be observed that the gel-breaking speed increases with the increasing temperature. This can be attributed to the lower strength of the UF capsule coat at higher temperatures, which accelerates the release of the SA capsule core and decreases gel viscosity. As can be observed in Figure 7a, the gel-breaking time for group A was determined to be 7, 6, and 5 days at 120 °C, 130 °C, and 140 °C, respectively. For group B (Figure 7b), the gel-breaking time was found to be 11, 10, and 9 days at 120 °C, 130 °C, and 140 °C, respectively. For group C (Figure 7c), the gel viscosity decreases slowly with well-maintained gel strength. The higher the dosage of the encapsulated breaker, the shorter the gel-breaking time and the better the gel-breaking performance. In addition, at the same time point, the gel viscosity is lower at higher temperatures.

#### 2.4.2. Effect of Encapsulated Breaker Dose on Gel-Breaking Performance

The dosage of the encapsulated breaker significantly affects the gel-breaking time and performance, such that a large dose will lead to premature gel-breaking and affect the construction results, while a small dose will lead to a longer gel-breaking time and possibly inhibit the process [18]. From Figure 7a,b, it can be concluded that at the same temperature, the larger the encapsulated breaker dosage, the faster the rate of decrease in gel viscosity. As a result, the gel in group A breaks approximately 4 days earlier than that in group B. Figure 8a shows the gel in group C, which maintains good elasticity upon extrusion. In comparison, the gel with the encapsulated breaker is more brittle, less elastic, and breaks into lumps upon extrusion (Figure 8b). The gels after extrusion can be seen in Figure 8c. The gel structure in the absence of the encapsulated breakers did not show much damage and could be restored to its original shape after extrusion, while the gel with the encapsulated breakers exhibited a decrease in gel structure strength due to the release of the breaker upon capsule rupture.

#### 2.4.3. Simulated Core Breakthrough Pressure Test

The core was removed from the core holder, and the gel broke into fragments in the core and formed a pathway (Figure 9). As shown in the breakthrough pressure curves in Figure 10, the gel breakthrough pressure can reach a maximum of 5 MPa; the breakthrough pressure of the gel with 2% encapsulated breaker rapidly decreases during days 5–6, while that of the gel with 1% encapsulated breaker decreases rapidly during days 9–10. The rapid decrease in the breakthrough pressure is due to the accumulated release of the encapsulated breaker over time, which destroys most of the gel structure, and as a result, the gel cannot withstand the applied pressure. The higher the content of the encapsulated breaker in the gel solution, the shorter the time required to break the gel structure. However, if the dosage is too small, the accumulated release of the encapsulated breaker will not be sufficient to break the gel structure and will not result in rapid pressure release.

## 3. Conclusions

An encapsulated breaker was prepared via in situ polymerization with UF resin as the capsule coat and SA as the capsule core. It can withstand temperatures of up to 120–140 °C and can be used for polymer gel temporary plugging. From experimentation, the optimal preparation conditions were determined to be a n_urea_:n_formaldehyde_ ratio of 1:1.8, a pH of 8, a temperature of 75 °C, and the utilization of Span 80/SDBS as the compound emulsifier.

A microstructure analysis of the UF/SA encapsulated breaker shows that the UF prepolymer is deposited on the surface of the capsule core, forming a smooth spherical body. The IR spectroscopy analysis indicates that SA had been successfully enclosed by UF, and the TG analysis confirms that the UF/SA encapsulated breaker can be safely stored at room temperature and is feasible for gel-breaking construction at 120–140 °C in the formation.

The encapsulation rate of the encapsulated breaker prepared with the Span 80/SDBS compound emulsifier is higher than that of those prepared with single emulsifiers. The conductivity test indicates that the encapsulated breaker is released via osmotic release. The higher the temperature, the lower the strength of the capsule coat, and the faster the release rate of the capsule core.

The effect of temperature and encapsulated breaker dosage on the gel-breaking performance was evaluated by measuring the viscosity change using a rheometer. At 120 °C, the gel with 2% encapsulated breaker can delay gel breaking for 6 days, while the addition of 1% can delay gel breaking for 11 days. At 130 °C, the gel with 2% encapsulated breaker can delay the gel breaking for 5 days, while that with 1% can delay gel breaking for 10 days. At 140 °C, gel breaking is delayed for 4 and 9 days with the addition of 2% and 1% encapsulated breaker, respectively. Conventional gel breakers can only delay gel breaking for 3~4 days. In summary, the encapsulated breaker was found to exhibit superior delayed gel-breaking performance in the 120–140 °C temperature range.

The results of the breakthrough pressure test in the core show that the slow-release UF/SA encapsulated breaker needs to accumulate to a certain level before release to exceed the breakthrough pressure of the gel. The higher the encapsulated breaker content in the gel solution, the shorter the time required to break the gel structure. However, if the encapsulated breaker content in the gel solution is too low, it will not cause a rapid release of pressure.

## 4. Materials and Methods

### 4.1. Instruments

HW-1 Electrothermal Constant Temperature Water Bath, Shandong Longkou Xianke Instrument Co., Ltd., Longkou, China; PHSJ-6L pH Meter, Shanghai INESA Scientific Instrument Co., Ltd., Shanghai, China; BGRL-2 Roller Heating Furnace, Qingdao Tongchun Petroleum Instrument Co., Ltd., Qingdao, China; IRAffinity-1S Fourier Infrared Spectrometer, Thermo Fisher Scientific (China) Co., Ltd., Beijing, China; TGA2 Thermogravimetric Analyzer, Mettler Toledo Technology (China) Co., Ltd., Beijing, China; Remagnet DDS-11A Conductivity Meter, Shanghai Yidian Scientific Instruments Co., Ltd., Shanghai, China; TY-2 type gripper, Nantong New Huacheng Research Instruments Co., Ltd., Nantong, China.

### 4.2. Materials

Formaldehyde Solution (HCHO, 37 wt%), Urea (H_2_NCONH_2_, 99%), Triethanolamine (C_6_H_15_NO_3_, 99 wt%), Sulfamic acid (SA, 99.5 wt%), Sodium dodecylbenzene sulfonate (SDBS, 95 wt%), Ammonium chloride (NH_4_Cl, 99.5 wt%), and Ethanol anhydrous (C_2_H_6_O, 99.5 wt%) were purchased from Aladdin Industrial Co., Ltd., Shanghai, China. Acrylamide (AM, 99 wt%), n-Octanol (C_8_H_18_O, 99 wt%), N,N-methylene bisacrylamide (C_7_H_10_N_2_O_2_, 99 wt%), and Span 80 were purchased from Sinopharm Chemical Reagent Co., Ltd., Beijing, China. Deionized water was self-made in the laboratory.

### 4.3. Methods

#### 4.3.1. Preparation of Encapsulated Breaker

Preparation of UF prepolymer

The urea to formaldehyde molar ratio (n_urea_:n_formaldehyde_) of 1:1.8 was used for the preparation of the UF prepolymer. Accordingly, a certain amount of formaldehyde solution with a mass fraction of 37% was weighed and placed in a beaker, and urea was added in two parts into the solution. The mixture was stirred to dissolve urea, and the mixed solution was transferred to a three-neck flask and placed in a 75 °C water bath. The pH of the solution was adjusted to 8 using triethanolamine during the heating process, and the solution was stirred at 800 r/min for 45 min. The resulting, slightly viscous, translucent UF prepolymer solution was then left to cool for 30 min.

Preparation of SA capsule core emulsion

The emulsifier is important for maintaining the dispersion and stability of the capsule core [19]. When Span 80 is used as a single emulsifier, the system stability is poor. Through comparison experiments, Span 80 and SDBS are preferably selected as the compound emulsifier. It was found that Span 80 as a single emulsifier gives rise to poor system stability. However, through comparison experiments, a mixture of Span 80 and SDBS as the compound emulsifier was preferable selected. Specifically, a certain amount of SA was slowly added to a beaker containing the compound emulsifier. The mixture was then stirred at 500 r/min for 30 min to yield a stable SA capsule core emulsion.

Preparation of UF/SA encapsulated breaker

Urea-formaldehyde resin has a low cost, simple production, good plasticity, good temperature resistance, and no reaction with sulfamic acid. Sulfonic acid can be used to remove the blockage of the oil layer and improve the permeability of the oil layer. Sulfonic acid has good temperature resistance. This combination has better adaptability than others and will not pollute the stratum. The capsule was prepared by enclosing the capsule core with the insoluble polymer produced from the polymerization reaction; this was achieved via in situ polymerization, with the prepolymer solution as the aqueous phase and the capsule core material as the oil phase in Figure 11 [20]. First, the SA capsule core emulsion was transferred to a three-neck flask. The flask was then placed into a 60 °C water bath and stirred. Afterwards, the prepared UF prepolymer solution was added to the flask, followed by an appropriate amount of n-octanol, which was added dropwise. Octanol acts as a defoamer, i.e., it can reduce the surface tension of the UF prepolymer, allowing it to settle on the SA capsule core surface faster, subsequently improving the encapsulation rate [21]. Then, an appropriate amount of 0.1 g/mL NH_4_Cl solution was added dropwise to the flask to lower the pH of the solution, and the flask was then stirred for an additional 2 h. At the end of the reaction, the product was washed with deionized water and anhydrous ethanol several times to remove impurities, before being dried in an oven at 60 °C to yield the UF/SA encapsulated breaker. The preparation process is shown in Figure 12. The temperature resistance of the encapsulated breaker can be improved by increasing the capsule coat thickness [22]. The encapsulation rate is related to the formulation of the UF prepolymer solution and the type of emulsifier used.

#### 4.3.2. Test and Characterization

Microstructure of the encapsulated breaker

Transmission electron microscopy (TEM) was used on SA capsule core emulsions formulated with different emulsifiers to analyze their dispersions, while thermal field emission scanning electron microscopy (SEM) was used to study the surface morphology of the encapsulated breaker. A certain amount of sample was dispersed in hexane, followed by metal spraying before SEM scanning [23]. The infrared (IR) spectrum of the UF/SA encapsulated breaker was recorded using a Fourier-transform infrared (FT-IR) spectrometer to analyze the components of the encapsulated breaker. Additionally, the thermal stability of the UF/SA encapsulated breaker was tested in the 0–600 °C range using a thermogravimetric (TG) analyzer with a heating rate of 10 °C/min.

Determination of the encapsulation rate

The encapsulation rate is used to determine whether the UF/SA encapsulated breaker can be successfully prepared via in situ polymerization, i.e., it indicates whether the water-soluble SA capsule core can be enclosed by UF resin [24]. An electronic balance was used to weigh an appropriate amount of the encapsulated breaker, which was recorded as W_i_. After extraction with xylene, the mixture was then centrifuged at 15,000 r/min for 15 min to separate the capsule coat and capsule core completely. The precipitate was then washed with anhydrous ethanol and dried in an oven at 60 °C. The product was then weighed and its weight was recorded as W_j_. The encapsulation rate, E_m_, was then calculated using Equation (1):(1)$$E_{m}=\frac{W_{j}}{W_{i}}\times 100\%$$

Electrical conductivity determination of the encapsulated breaker

The dissolution of the capsule core SA in water leads to changes in the electrical conductivity of the solution, and the release amount and rate of SA can be determined using the conductivity method [25]. First, three encapsulated breaker samples and three SA samples of the same mass were weighed, with the SA samples used as controls. The samples were then transferred into separate dropping bottles containing 100 mL of deionized water. The measuring electrode was then placed into the dropping bottle and the solution conductivity was recorded at different times at 120 °C, 130 °C, and 140 °C.

#### 4.3.3. Gel-Breaking Performance Test of the Encapsulated Breaker

First, 300 mL of gel solution containing monomer AM, crosslinker N-MAM, and heat resistant material was prepared and divided equally into 3 groups: A, B, and C [26]. Then, 2 g and 1 g of the UF/SA encapsulated breaker was added to groups A and B, respectively. Group C served as the control group. The groups were then individually mixed and further divided into three more parts for gel formation. The gels were then subjected to gel-breaking tests conducted at 120 °C, 130 °C, and 140 °C. The gel-breaking performance of the encapsulated breaker was evaluated based on gel-breaking time and gel viscosity, i.e., [27], complete gel-breaking is achieved when the gel viscosity is lower than 5 mPa s [28].

Simulated core experiments are required to further investigate the gel-breaking performance of the UF/SA encapsulated breaker, and to verify whether the gel can exert effective plugging within a certain period of time [29]. The gel solutions with 2% and 1% encapsulated breaker were, respectively, injected into the artificial core and heated to 130 °C to test the breakthrough pressures of the gels at different times. The breakthrough pressure is related to the gel strength, i.e., the higher the strength, the higher the breakthrough pressure [30]. In the laboratory, breakthrough pressure testing is a mature method with no potential limitations. Therefore, testing the breakthrough pressures of gels with different contents of the encapsulated breaker can reflect the gel-breaking performance of the encapsulated breaker.

## Figures and Tables

**Figure 1 gels-09-00387-f001:**
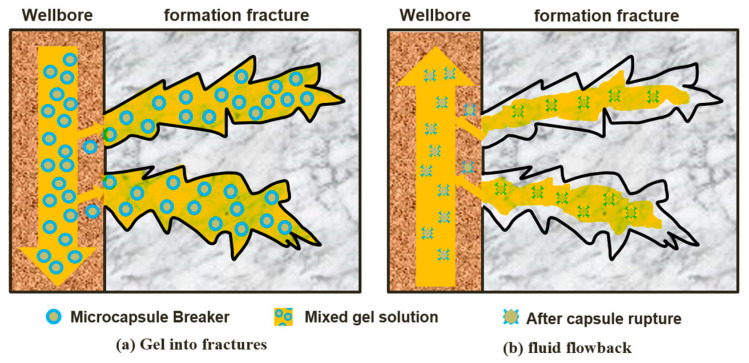
Schematic construction diagram of encapsulated breaker, (**a**) Gel into fractures, (**b**) fluid flowback.

**Figure 2 gels-09-00387-f002:**
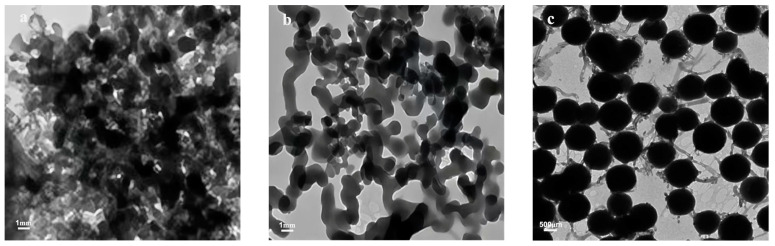
TEM images of capsule core emulsions prepared with (**a**) Span 80, (**b**) SDBS, and (**c**) compound emulsifier.

**Figure 3 gels-09-00387-f003:**
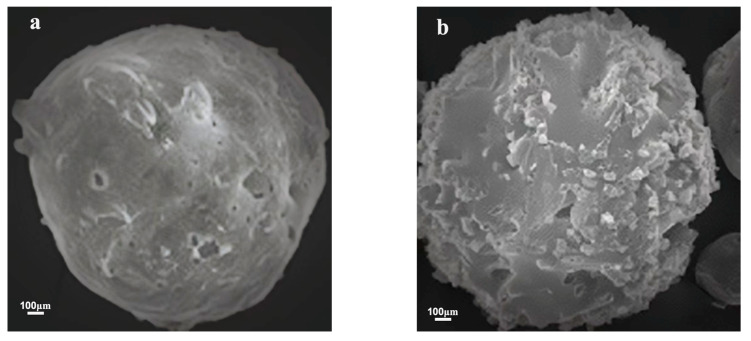
SEM images of encapsulated breakers prepared with (**a**) n_urea_:n_formaldehyde_ = 1:1.8, (**b**) n_urea_:n_formaldehyde_ = 1:1.7.

**Figure 4 gels-09-00387-f004:**
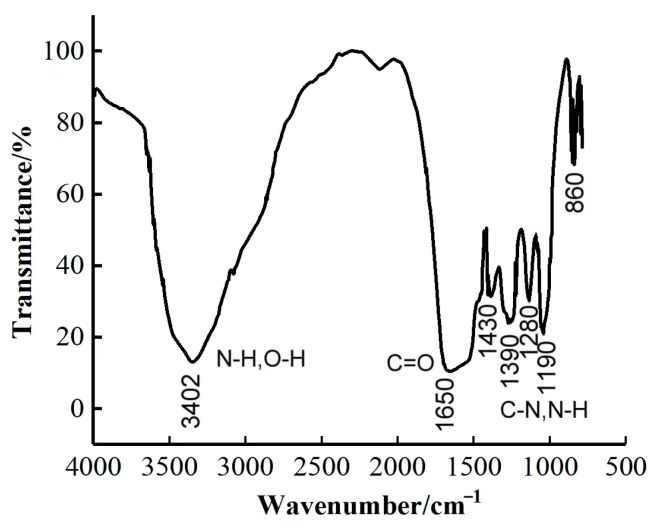
IR spectrum of the encapsulated breaker.

**Figure 5 gels-09-00387-f005:**
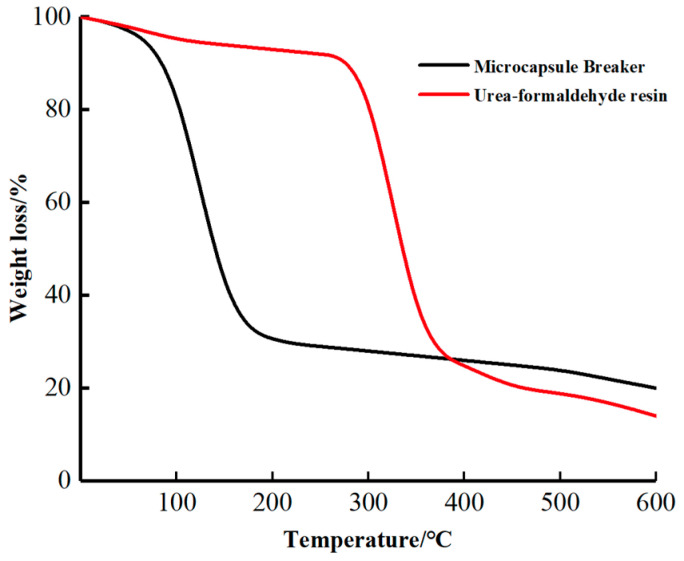
TG curves of the encapsulated breaker and the UF resin.

**Figure 6 gels-09-00387-f006:**
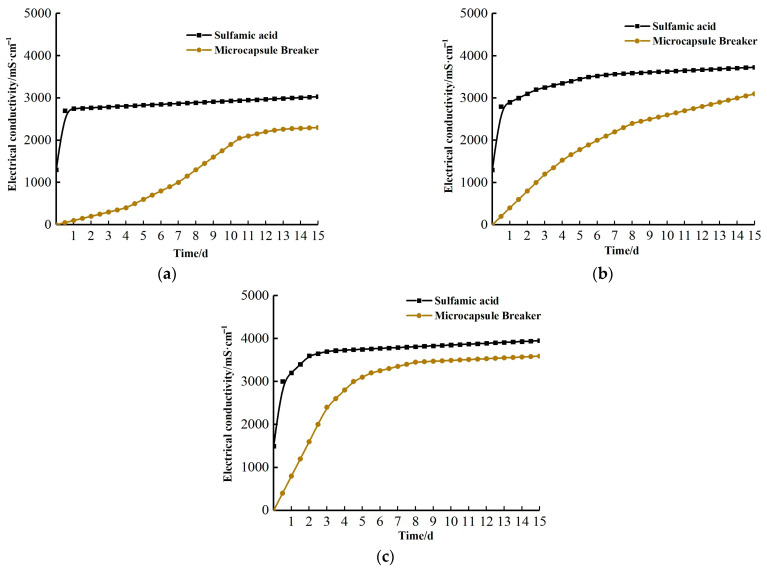
Solution conductivity at different temperature conditions. (**a**) Temperature 120 °C. (**b**) Temperature 130 °C. (**c**) Temperature 140 °C.

**Figure 7 gels-09-00387-f007:**
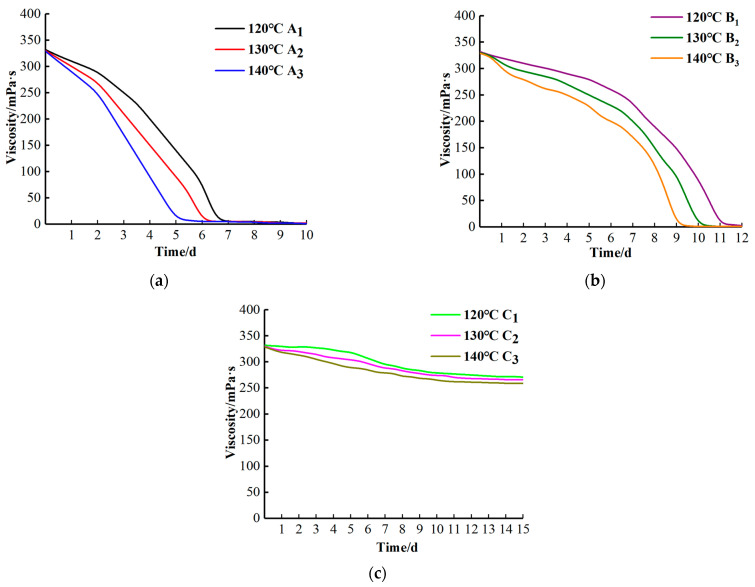
Variation of viscosity versus time under different temperature conditions. (**a**) Gel with 2% UF/SA encapsulated breaker. (**b**) Gel with 1% UF/SA encapsulated breaker. (**c**) Gel without UF/SA encapsulated breaker.

**Figure 8 gels-09-00387-f008:**
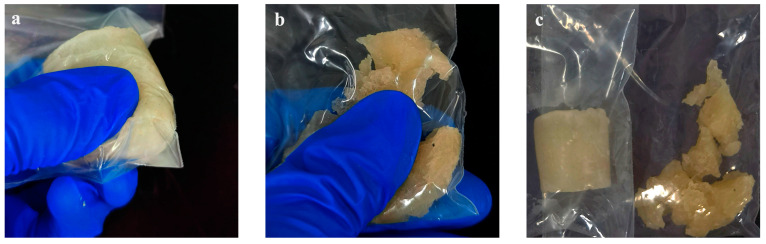
Effect of encapsulated breaker addition on gel structure. (**a**) The gel without the encapsulated breaker. (**b**) The gel with the encapsulated breaker. (**c**) Comparison.

**Figure 9 gels-09-00387-f009:**
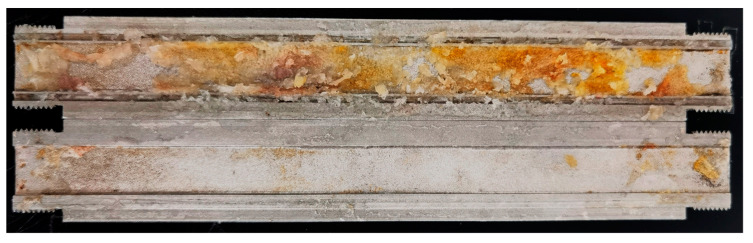
Gel state after simulated core experiment.

**Figure 10 gels-09-00387-f010:**
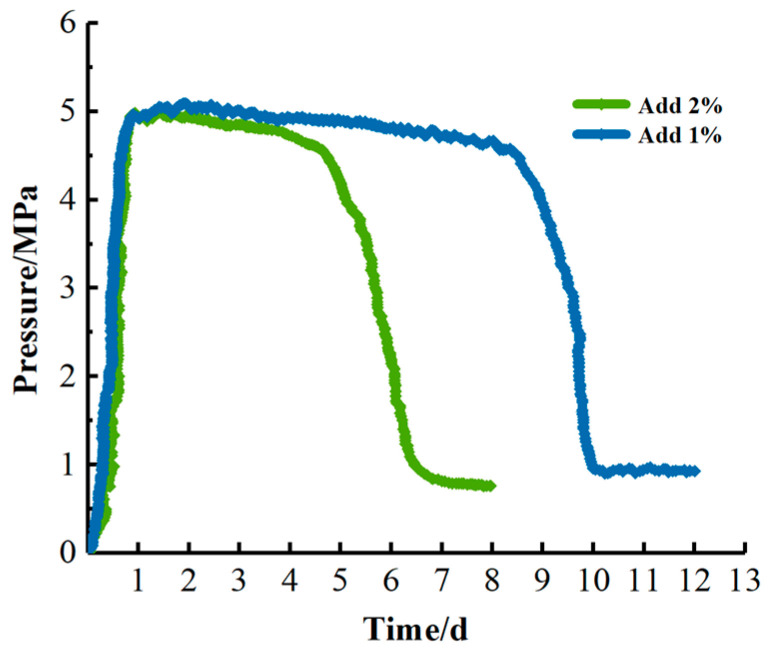
Breakthrough pressure curve in the core.

**Figure 11 gels-09-00387-f011:**
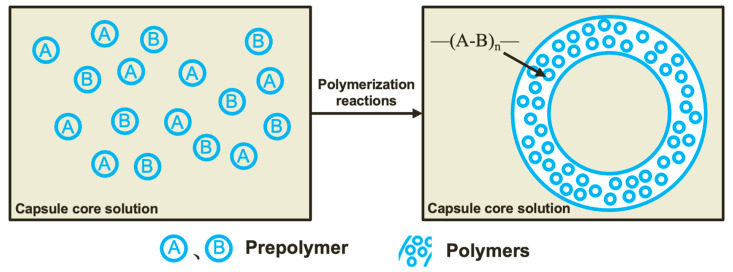
Schematic diagram of encapsulated breaker preparation via in situ polymerization.

**Figure 12 gels-09-00387-f012:**
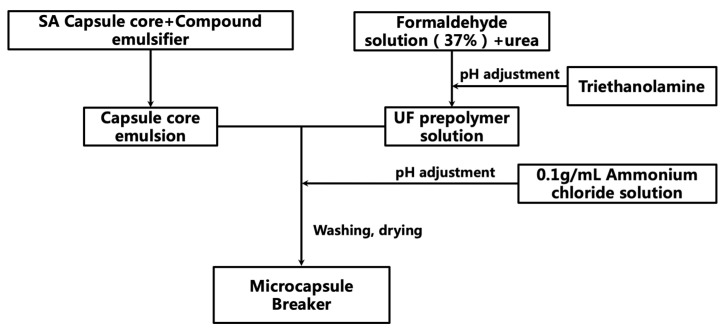
Preparation flow chart of the UF/SA encapsulated breaker.

**Table 1 gels-09-00387-t001:** Encapsulation rates of capsule breakers prepared under different conditions.

Group	Molar Ratio (n_urea_:n_formaldehyde_)	Emulsifier	Encapsulation Rate/%
1	1:1.7	Span 80	68.32
2	1:1.7	SDBS	70.64
3	1:1.7	Span 80/SDBS	79.47
4	1:1.8	Span 80	74.38
5	1:1.8	SDBS	77.52
6	1:1.8	Span 80/SDBS	85.86

## Data Availability

Not applicable.

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
