# Peer review of "Preparation of Encapsulated Breakers for Polymer Gels and Evaluation of Their Properties"

_gels, 2023, doi:10.3390/gels9050387_

Round 1

Reviewer 1 Report

The study investigates a new type of gel breaker to address the problem of premature reduction in gel viscosity at high temperatures. The encapsulated polymer gel breaker, consisting of a urea-formaldehyde (UF) resin as the capsule coat and sulfamic acid (SA) as the capsule core, was successfully encapsulated via in situ polymerization. The encapsulated breaker showed slow-release properties and could withstand temperatures up to 120-140 °C. The study also identified optimal preparation conditions, which resulted in significantly improved gel-breaking performance, with gel-breaking being delayed for nine days at 130 °C. The study is well-structured and provides a comprehensive characterization of the encapsulated breaker, making it a valuable contribution to the field of gel breakers. The study is a novel contribution to the domain of gel breakers. However, there are major comments relevant to this study that are necessary to be addressed before its acceptance for publication:

1.             What specific advances or improvements does this study offer over existing encapsulated breaker technology, if any? This is to expand the Introduction section, which seems too thin for this study.

2.             How significant is the improvement in performance of the encapsulated breaker compared to alternatives? Explain in detail.

3.             What is the rationale for selecting urea-formaldehyde resin as the capsule coat and sulfamic acid as the capsule core, and how does this combination compare to other potential encapsulation materials?

4.             Are the optimal preparation conditions identified in the study feasible and practical for industrial-scale production?

5.             How accurately do the simulated core experiments reflect real-world conditions, and what are the potential limitations of this approach?

6.             Is the use of the Span 80/SDBS compound emulsifier common or well-established in the field, or is this a novel approach?

7.             Are there any potential safety or environmental concerns associated with the UF/SA encapsulated breaker that should be addressed?

8.             What is the potential economic impact of the UF/SA encapsulated breaker compared to other gel breakers in terms of cost, efficiency, and sustainability?

9.             Are the results of the breakthrough pressure test sufficiently conclusive, and are there any potential limitations to this approach?

10.          How well does the UF/SA encapsulated breaker perform in different types of formations, and what are the potential limitations or challenges in using this technology in different geological settings?

11.          How did the selection of emulsifiers (SDBS, Span 80, and Span 80/SDBS compound emulsifier) impact the dispersion and stability of the encapsulated breakers, as observed in the TEM characterization of capsule core emulsions?

12.          What is the relationship between the ratio of nurea:nformaldehyde and the microstructure of the encapsulated breaker, as seen in the SEM images?

13.          How was the formation of the UF resin capsule coat around the SA capsule core confirmed through the IR spectroscopy analysis?

14.          Can the TG analysis results be used to determine the optimal thickness of the capsule coat to improve the temperature resistance of the encapsulated breaker?

15.          How does the encapsulation rate of the encapsulated breaker vary with the emulsifier type used during the preparation process, as observed in Table 1?

16.          Can the differences in electrical conductivity between the SA solution and the encapsulated breaker solution, as seen in Figure 6, be used to determine the extent of damage to the encapsulated breaker?

17.          What is the relationship between temperature and the gel-breaking time of the encapsulated breaker, as shown in Figure 7?

18.          How does the dosage of encapsulated breaker affect the gel-breaking time and performance, as observed in Figures 7 and 8?

19.          How does the encapsulated breaker affect the gel structure, as shown in Figure 8?

20.          How does the simulated core breakthrough pressure test show the effect of the encapsulated breaker on the gel structure and stability, as shown in Figure 9 and Figure 10?

21.          What is the significance of the UF resin capsule coat in enabling the slow release of the SA capsule core, as observed in the electrical conductivity results?

22.          Can the encapsulated breaker be used effectively for polymer gel temporary plugging at temperatures up to 120-140°C, and how does its delayed gel-breaking performance compare to conventional gel breakers?

Texts are not clear in some places and punctuation marks are missing.

Reviewer 2 Report

This interesting paper is regrettably written in a careless way and thus, needs, a major, mandatory revision.

1. What is statistical scattering of the encapsulation rate supplied in Table 1? No scientific paper should be published without accurate statistical scattering of the reported results. Are all of the four figures of the encapsulation rate, reported in Table 1, significant?

2. Recent results should be referred, see:

Roy P. K. et al., Manufacture and properties of composite liquid marbles, Journal of Colloid and Interface Science, 575 (2020) 35-41.

3. Scale bar in Figure 2 is unclear.

4. Figures 8-9 need scale bars.

5. The accuracy of the temperature control should be supplied in the revised version of the manuscript.

6. The accuracy of the TG test should be supplied in the revised version of the manuscript.

7. The accuracy of PH measurement should be supplied in the revision.

8. The English should be edited. What is "Conductivity determination of the encapsulated breaker"? 

9. The novelty of the reported approach should be clearly stated.

The English should be edited.

Reviewer 3 Report

In my view, the authors have conducted an interesting work which has good impact to the field. Regardless, the authors should carefully address the comments shown below before I recommend this work for publication:

(1) With regards to Figure 1: if the authors need to provide more detailed analysis of the SEM images of the capsules with the literature. 

(2) What parameters regulate the formation and structure of these capsules? Please add more details. 

(3) With regards to IR Spectroscopy data shown in Figure 4, please provide references for the assigned peaks. Additionally, mark the main peaks along with the corresponding bond in the figure itself. 

(4) What dictates the concentration of emulsifiers used in this study? Why this range was selected? Add a high level information so the readers can use this work in the future. 

(5) What are the main assumptions used to calculate the encapsulation rate? Please add the error bar for the estimated rate in Table 1. 

(6) Please clarify the caption of Figure 8. Please add more details in the caption in terms of what these images are indicating. 

(7) State the novelty of the work clearly and provide recommendations for future research directions. 

(8) Improve the figure quality and proofread the manuscript to ensure there are no grammatical as well as typographical errors.

(9) The authors need to improve the literature review. There are studies that highlight the importance of methods used for encapsulation, emulsification and rheology on microstructure of material. 

1. https://doi.org/10.1007/s42452-020-03879-5

2.https://doi.org/10.1021/acsanm.3c00529

3.  https://doi.org/10.1016/j.cofs.2023.101003

The English can be improved. There are a few typos and in some places the grammatical errors can be observed.  

Round 2

Reviewer 1 Report

The reviewer cannot identify changes implemented in the revised manuscript for the suggestions to the first version.

There are no major comments.

Reviewer 2 Report

The paper is publishable.

The English is OK.

Reviewer 3 Report

The authors have improved the manuscript to a good extent. There are a few minor comments to address:

(i) The references should be fixed. A few of the references have been removed and they need to be considered. 

(ii) Proofread the manuscript. 

Manuscript needs to be proofread.